



# Seasonal Sea Ice Prediction with the CICE Model and Positive Impact of CryoSat-2 Ice Thickness Initialization

Shan Sun[1] and Amy Solomon[2]

[1]NOAA/Global Systems Laboratory
[2]NOAA/Physical Sciences Laboratory and University of Colorado Boulder, Cooperative Institute for Research in Environmental Sciences

**Correspondence:** Shan.Sun@noaa.gov

**Abstract.** The Los Alamos sea ice model (CICE) is being tested in standalone mode for its suitability for seasonal time scale prediction. The prescribed atmospheric forcings to drive the model are from the NCEP Climate Forecast System Reanalysis (CFSR). A built-in mixed layer ocean model in CICE is used. Initial conditions for the sea ice and the mixed layer ocean in the control experiments are also from CFSR. The simulated sea ice extent in the Arctic in control experiments is generally in good

agreement with observations in the warm season at all lead times up to 12 months, suggesting that CICE is able to provide useful ice edge information for seasonal prediction. However, the ice thickness forecast has a positive bias stemming from the initial conditions and often persists for more than a season, limiting the model's seasonal forecast skill. In addition, thicker ice has a lower melting rate in the warm season, both at the bottom and top, contributing to this positive bias. When this bias is removed by initializing the model using ice thickness data from satellite observations while keeping all other initial fields

unchanged, both simulated ice edge and thickness improve. This indicates the important role of ice thickness initialization in sea ice seasonal prediction.

## 1   Introduction

Sea ice concentration observations from passive microwave satellites show that ice coverage has decreased rapidly in the Arctic in recent decades, and in the Antarctic in recent years, the two regions with greatest warming on earth in recent decades

(Chapman and Walsh, 2003). Global climate models (e.g., IPCC, 2014, 2021) suggest that further reduction in sea ice coverage and thickness will occur in the coming decades. Sea ice in the polar oceans has a major impact not only on the regional energy balance but on the global climate as well. A relatively thin material layer between the atmosphere and ocean, sea ice works to regulate the fluxes between the two components and amplifies climate feedbacks as sea ice has a higher albedo than open water  (e.g., Holland and Bitz, 2003; ACIA, 2005; Dethloff et al., 2006). Thus, change in sea ice is one of the most sensitive

and visible indicators of our changing climate. Reliable sea ice prediction is important not only for forecasting in the polar regions but also is expected to improve predictability at mid-latitudes at subseasonal to seasonal (S2S) time scales due to teleconnections (e.g., Randall et al., 1998; Jaiser et al., 2012; Li et al., 2014).

Finding physical processes or anomalies leading to predictability at S2S time scales is a challenging research topic. Sea ice is found to be one candidate due to the persistence of seasonal anomalies  (Blanchard-Wrigglesworth et al., 2011a; Holland et al.,





2011; Bushuk et al., 2019). Progress has been made on sea ice modeling in recent decades, which holds promise for improving both medium range and climate predictions (e.g., Wang et al., 2013; Hebert et al., 2015). It has been shown that weather predictions from a numerical model incorporating a sea ice model have higher skill than those based on persistence of Arctic sea ice (Grumbine, 2003; Hebert et al., 2015; Intrieri et al., 2020). The predictability skill is found to rely on the accuracy of the sea ice initial conditions (Holland et al., 2011; Blanchard-Wrigglesworth et al., 2011b; Wang et al., 2013). In particular,

studies have found improvement of sea ice forecasts at seasonal to interannual timescales with a more realistic sea ice thickness initialization, either observed or approximated, as sea ice volume is more persistence than sea ice coverage (Krinner et al., 2010; Chevallier and Salas-Mélia, 2012; Day et al., 2014a; Allard et al., 2018; Blockley and Peterson, 2018; Schröder et al., 2019, to name a few). It has also been shown that sea ice predictability is season-dependent (Holland et al., 2011; Day et al., 2014b; Bushuk et al., 2020) and predictability of minimum sea ice extent depends on atmospheric spring and early summer conditions,

where inclusion of melt ponds is seen to make a difference (Liu et al., 2015; Schröder et al., 2019; Bushuk et al., 2020).

A fully coupled atmosphere-ocean-sea ice model is considered the ultimate tool for sea ice prediction at seasonal and climate time scales (e.g., IPCC, 2014, 2021). In order to separate various feedbacks among the components of a fully coupled model, in this study we aim to validate the sea ice model in a standalone mode for seasonal prediction with prescribed atmospheric forcings. We chose the Los Alamos Community Ice CodE (CICE; Hunke et al., 2015) as the sea ice model in this study, due

both to its popularity in the community (e.g., Shaffrey et al., 2009; Holland et al., 2011; Roberts et al., 2015; Metzger et al., 2014; Intrieri et al., 2020), and the fact that it is being incorporated into NOAA's Unified Forecast System (UFS). We hope to build a set of baseline results for future studies including coupled models using the same sea ice module. We also investigate the sensitivity of prediction skill of the standalone CICE when initialized with different ice thickness products, as suggested in various studies mentioned earlier, in order to be close to the potential predictability (Palmer, 2006), where the model is

assumed to be perfect and the only source of error arises from the uncertainties in the initial conditions. The detailed model setup is described in section 2. Its basin-wide and regional performance at different lead times, as well as different initialization dates, are presented in section 3, followed by a discussion in section 4.

## 2   Model Setup

CICE is a dynamic–thermodynamic sea ice model designed for application in a global climate model. We adopted the CICE

code V5.1.2 (Hunke et al., 2015) and performed simulations in the global domain. The linear function of salinity (ktherm=1) was used for the freezing temperature. Mushy physics and elastic-anisotropic-plastic rheology were specified. Since CICE needs to operate with an SST which can be either prescribed or generated by its own built-in mixed layer model, we chose the latter in this study for the sake of consistency between the SST and the ice state. A combination of an Arctic bipolar grid projection and a Mercator projection of the rest of the globe (Fig.1 in Bleck and Sun, 2004) is used to carry out all CICE

experiments, where the horizontal resolution is 15 km at the North Pole and 30 km at 60°N and 60°S, with a bathymetry interpolated from ETOPO1 data (Amante and Eakins, 2009).





## 2.1 Atmospheric Boundary Conditions

The atmospheric boundary forcings are based on the 6 hourly archives from CFSR[1] (Saha et al., 2010). They include downward surface radiation of both short- and longwave, 10 m wind, 2 m temperature, 2 m specific humidity, and precipitation (broken down into rain and snow) at 0.2° resolution from 2011 to 2017. These fields were interpolated horizontally onto the model grid specified earlier in the pre-processing step.

## 2.2 Initial Conditions

In the control experiments, CFSR reanalysis data were used as initial conditions for both the ice state and the ocean state. These include ice coverage and thickness at 0.2° resolution, and sea surface temperature and sea surface salinity at 0.5° resolution. All of these data were interpolated onto the compound model grid.

Furthermore, in order to take advantage of latest satellite observations and investigate the sensitivity of seasonal sea ice prediction to initial conditions, as seen in studies mentioned earlier, additional CICE experiments were carried out by initial-izing from satellite observations of CryoSat-2 (Grosfeld et al., 2016), which provides monthly mean estimates of the Arctic ice thickness during boreal winter (October to April) since 2011. We use this data to replace the ice thickness from CFSR in the initial conditions. Since CryoSat-2 data are only available in the Arctic, the initial Antarctic sea ice thickness still comes from CFSR. For grid cells starting with zero ice thickness in the CryoSat-2 data, the sea ice concentration from CFSR is reset to zero initially to be consistent. The remaining initial conditions as well as the atmospheric boundary conditions are the same as in the control experiments. This set of experiments with an alternative CryoSat-2 ice thickness initialization is called the "alt-init" experiments.

The control experiments were initialized at the beginning of each month from April 2011 to December 2017. The alt-init experiments initialized with the CryoSat-2 Arctic ice thickness were carried out monthly in the winter season (October to April) when this data was available, as shown in Table 1. The integration time was 12 months throughout all model runs.

## 3 Model Results and Verification

All comparisons and verifications in this section are carried out on the native grid, except for overall evaluations at the hemi-spheric or regional scales.

## 3.1 Hemispheric Scales

The simulated sea ice extent (SIE) and sea ice volume (SIV) in the Arctic and Antarctic in the control experiments are shown in Fig. 1. Each solid line of different color represents one 12-month integration, where the initial time is marked by squares. The 12 once-per-month runs in 2014 are shown here as an example, as year to year variations are relatively small. Circle symbols represent SIE observations from NSIDC (Meier et al., 2012) and SIV observations from CryoSat-2, respectively. Also shown

[1] available at http://rda.ucar.edu/datasets/ds094.0



**Table 1.** Details in two sets of CICE experiments.

| Experiments | Control Runs | CryoSat-2-based ("alt-init") Runs |
|---|---|---|
| Atmospheric boundary conditions | CFSR | CFSR |
| Initial conditions for ice and ocean | CFSR | Same as in the control, except initializing the Arctic with CryoSat-2 ice thickness |
| Initialization Months | Apr 2011 – Dec 2017 | Oct-Apr 2011-2017 |

in "x" symbols are SIV estimates from Pan-Arctic Ice Ocean Modeling and Assimilation System (PIOMAS, Schweiger et al., 2011). The Arctic SIE forecast matches observations better in the warm season than in the cold season at all lead times, and a positive SIE bias is seen in the cold season. When the SIV in the Arctic is higher than observations or reanalysis to begin with, this positive SIV bias often remains in the model throughout the forecast. In the Antarctic, a positive SIE bias is seen in almost

all seasons in the seasonal forecast. There is no SIV observation available in the Antarctic at the moment.

To quantify the forecast skill, root mean square error (RMSE) of the Arctic SIE for each target month at lead times up to 12 months is shown in the upper panel of Fig. 2 for the control (left) and the alt-init (right) runs, where white spaces indicate lack of CryoSat-2 data from May to September for model initialization. The RMSE is against NSIDC observations and is averaged over all years. The control runs did a credible job in forecasting SIE in summer and fall in the Arctic, at all lead times up to 12

months. The forecast skill for winter and early spring at lead times longer than 3 months is the poorest. Initializing the model with CryoSat-2 ice thickness helps reduce some error in the Arctic SIE seen in the control runs, indicating the impact of initial ice thickness on the SIE prediction.

The lower panel of Fig. 2 is similar to the upper one, except for SIV with respect to PIOMAS reanalysis. The control runs started with a SIV higher than PIOMAS estimates in all months, and maintained a positive SIV bias at different magnitudes for

all target months at all lead times. The most obvious bias of SIV is the "summer barrier" in Fig. 2(c), where the reemergence of skill at a longer lead time prior to the winter initialization can be seen clearly. This barrier as well as overall positive bias of SIV is much reduced in the alt-init runs with the CryoSat-2 ice thickness initialization in Fig. 2(d).

The RMSE of the Antarctic SIE is shown in Fig. 3 for the control experiments from all years. The model is able to make reliable SIE forecasts at all lead times for the austral winter, spring and especially fall. The lowest seasonal forecast skill

occurred in the austral summer, in contrast to the lowest skill in late winter in the Arctic. We are unable to evaluate the SIV skill in the Antarctic as there are no observations or reanalysis products available at this time.

### 3.1.1 Pan-Arctic forecasts at 6-month lead time

In order to see the evolution of sea ice distribution, the Arctic SIE in the control and the alt-init experiments is shown in Fig. 4 at the initial time of April 1 2016 and the 6-month forecast targeted in October 2016, as well as the corresponding AMSR2

satellite observations. The initial SIE in both experiments is very close to AMSR2. At month 6, comparing to AMSR2, a positive SIE bias appeared mostly in the Beaufort and Chukchi Seas in the control run, while SIE from the alt-init run is much



closer. One reason for this is the difference in the ice thickness shown in Fig. 5, where initial and 6-month forecast SIV are plotted for the control and the alt-init runs, together with the corresponding CryoSat-2 observations. The control run starts with ice much thicker than observed in the Beaufort, Chukchi and East Siberian Seas in spring, and the signature of this extra thick
ice can still be seen in fall 6 months later (lower left panel).

Even in the month of October when a relatively higher skill of Arctic SIE forecast is seen, thicker ice in the initial conditions in April appears to be one dominant source for the error in SIE and SIV in the 6 month forecast. Both SIE and SIV forecast improved when this bias was corrected in the alt-init run. The positive impact on the forecast skill from a realistic sea ice thickness initialization shown here is consistent with findings from  Day et al. (2014a); Allard et al. (2018); Blockley and Peterson
(2018); Schröder et al. (2019).

### 3.1.2   Pan-Arctic 12-month forecasts

Since all CICE model runs here are integrated for 12 months, Fig. 6 presents the modeled SIVs compared to CryoSat-2 observations initially in January 2016 and its 12 month forecast for December 2016, respectively, in both experiments. Similar conclusions can be drawn here as in Fig. 5, i.e., erroneous ice thickness seen in the Beaufort, Chukchi and East Siberian Seas
in the 12-month forecast in the control run can be traced back to its initial SIV bias, while the SIV forecast in the alt-init run is closer to but thinner than CryoSat-2.

To explore the mechanism behind this, Fig. 7(a) and (b) show the simulated Arctic SIE and SIV through the 12-month integration from the runs in Fig. 6 and their comparison to NSIDC and PIOMAS. Although both runs overestimated the SIE and SIV in spring, the SIE and SIV in the alt-init run ended up closer to NSIDC and PIOMAS, respectively, in fall than in the
control run. The transitions of SIE and SIV from spring to fall are different in these two runs, which can be seen through their tendencies averaged north of 67.5°N as shown in Fig. 7(c), where there are more negative thermodynamics-linked tendencies in both SIE and SIV in summer in the alt-init run than in the control run, and the dynamics-linked tendencies are similar in these two runs. Furthermore, Fig. 7(d) shows the melting tendencies averaged north of 67.5°N from the top, bottom and lateral, respectively, in both runs. The biggest difference occurred in the bottom melt and some difference in the top melt, while the
lateral melt is similar. It appears that the extra melt from the ice bottom and top in summer in the alt-init run leads to a closer match in SIE and SIV with observations in fall, and therefore the melt rate in the alt-init run is more adequate than in the control run.

To further assess why the melt is different in these two model runs, the ice thickness predicted for July 2016 with January 2016 initialization is shown in the upper panel in Fig. 8 for the control and alt-init experiments. The melt differences in the
alt-init and control runs are shown in the lower-left panel for the top melt and lower-right for the bottom melt. Apparently, there are more bottom and top melt in the Beaufort, Chukchi and East Siberian Seas in the alt-init run than in the control run, where the ice thickness in this region is mostly below 1.5 m in the alt-init run compared with above 1.5 m in the control run. This implies that relatively thinner ice in the alt-init run favors more melt both at the bottom and top than in the control run during the month of July, suggesting the importance of proper ice thickness in the melting process.





In summary, the Arctic SIE prediction in the control run with the CICE model has a higher skill in the warm season than
in the cold season at almost all lead times. However, the biggest Arctic SIV bias is in summer at 3 month lead time, and in
fall/winter at roughly 6/9 month lead times. This may be the result of a combination of thicker ice initialization in the cold
season and insufficient melt in the warm season. When the model is initialized with a more realistic thinner ice from CryoSat-2
observations in the alt-init runs, a higher forecast skill is achieved in both SIE and SIV at all lead times. Nevertheless, there is

still a positive SIV bias in late winter in the alt-init runs at 3- and 6-month lead times. More details can be seen in the regional
investigation next.

### 3.1.3    Regions

Analysis of the sea ice prediction skill is performed in regions defined in https://arctic-roos.org in Fig. 9. For the purpose
of this study, we refer to the combination of the Barents, Kara, and Greenland Seas as BKG seas. To investigate the season-

dependency of the forecast skills seen earlier, we present SIE forecasts in each basin in Fig. 10 in the control (solid blue)
and alt-init (dashed red) runs throughout the 12-month integration initialized in April of 2013 to 2016, as well as AMSR2
observations (circle). The control run has a reasonable SIE in all 8 basins for summer and fall, but it overpredicts SIE in the
BKG Seas, Baffin Bay and East Siberian Sea for winter at lead times longer than 6 months. When the run is initialized with a
more realistic SIV in the alt-init experiments, the SIE prediction is much improved in the East Siberian Sea at lead times above

6 months while it remains unchanged in the BKG seas and Baffin Bay.

        The SIV forecast from the same model runs in Fig. 10 is shown in Fig. 11 in each basin, together with the CryoSat-2
observations. The initial positive bias of SIV in the Beaufort, Chukchi, East Siberian and Laptev Seas seen in the control run is
mostly removed at almost all lead times in the alt-init runs, indicating that the positive SIV bias in the initialization is mainly
responsible for the positive bias in the seasonal forecast in the Beaufort, Chukchi, East Siberian and Laptev Seas. Apparently,

positive biases of SIE and SIV in the Baffin Bay and BKG Seas in winter come from a different source. There could be errors
in the simplified mixed layer ocean, atmospheric forcings, or the CICE model itself, most likely from the interaction with the
Atlantic ocean, which leads to the sea ice edge extending too far southward in winter. Narrowing down the possible sources of
this error needs additional experiments, which is beyond the scope of this study.

### 4    Discussion

In this study, we carry out numerical experiments to evaluate sea ice prediction skills at seasonal time scales in a standalone
CICE ice model incorporating a mixed layer ocean model. The control run is initialized with CFSR data and is subjected
to prescribed atmospheric forcing, also from CFSR. Multiple year-long experiments are initialized at monthly intervals. The
control run does a credible job in forecasting SIE both in the Arctic and Antarctic for most seasons at lead times up to 12
months. The lowest skill for SIE forecasts is in late winter and early spring in the Arctic, and in the austral summer in the

Antarctic.

We found one of the factors limiting the model forecast skill in the Arctic is the SIV bias in the initialization in the control run: the initial SIV from CFSR is consistently higher than the satellite observations from CryoSat-2. The excessive ice volume is retained in the rest of the model integration. We are able to remove this bias in the alt-init runs, when the CFSR ice thickness in the Arctic is replaced by the CryoSat-2 ice thickness in the initial conditions, keeping everything else unchanged. Although

the CryoSat-2 ice thickness data are only available from October to April in the Arctic, the multi-year runs initialized by CryoSat-2 ice thickness clearly show the reduced biases in SIE and SIV forecast at all lead times. In addition, comparison of different melting rates in these two runs revealed a key process, i.e., melting can be too weak at the ice bottom and top in the warm season when ice is thicker than a threshold (1.5 m in this case).

    There is still a positive bias at a smaller magnitude in both SIE and SIV in the alt-init experiments in the BKG Seas and Baffin

Bay, all in the vicinity of the Atlantic. The possible sources for this could be the simplified mixed layer ocean, atmospheric forcing, the CICE model itself, or the inherent limit to predictability. We are unable to identify it in this setup. It would be a subject in the future with additional experiments.

    As the sea ice module in the NOAA's UFS, the CICE model results presented here are consistent with those in the current NOAA operational CFSv2 (Wang et al., 2013), where the coupled model produced the atmospheric reanalysis that is used to

force the CICE runs here. The RMSE shown in Fig. 2 here is larger than in Wang et al. (2013) for a number of reasons: (1) the latter is from a fully coupled model allowing feedback between atmosphere, ocean and sea ice, while this study uses an uncoupled sea ice model without any feedback; (2) both the ocean model and the sea ice model are different; (3) the validation period is different. Wang et al. (2013) compared to the 26-year climatology of 1981-2007, while the 2011-2017 period used in this study is characterized by a lower SIE observation than decades earlier.

This study confirms the importance of the accuracy of initial ice thickness for seasonal sea ice prediction in a standalone ice model. It suggests that there exists a potentially important source of additional skill in the seasonal forecast in the initial ice thickness. Hence, assimilation of observed sea ice thickness appears to be highly relevant for advancing seasonal prediction skill.

*Author contributions.* SS and AS designed the study, performed the analyses and wrote the manuscript. SS carried out the numerical exper-

iments.

*Competing interests.* The authors declare that they have no conflicts of interest.

*Acknowledgements.* This research is funded by the Global Model Test Bed at Developmental Testbed Center under NGGPS. We thank David Bailey for helpful discussions and sharing a script to convert initial conditions to restart conditions for CICE, Xingren Wu for suggesting using a mixed layer ocean instead of SST from CFSv2 and Rainer Bleck for constructive suggestions during the interval review. Discussions





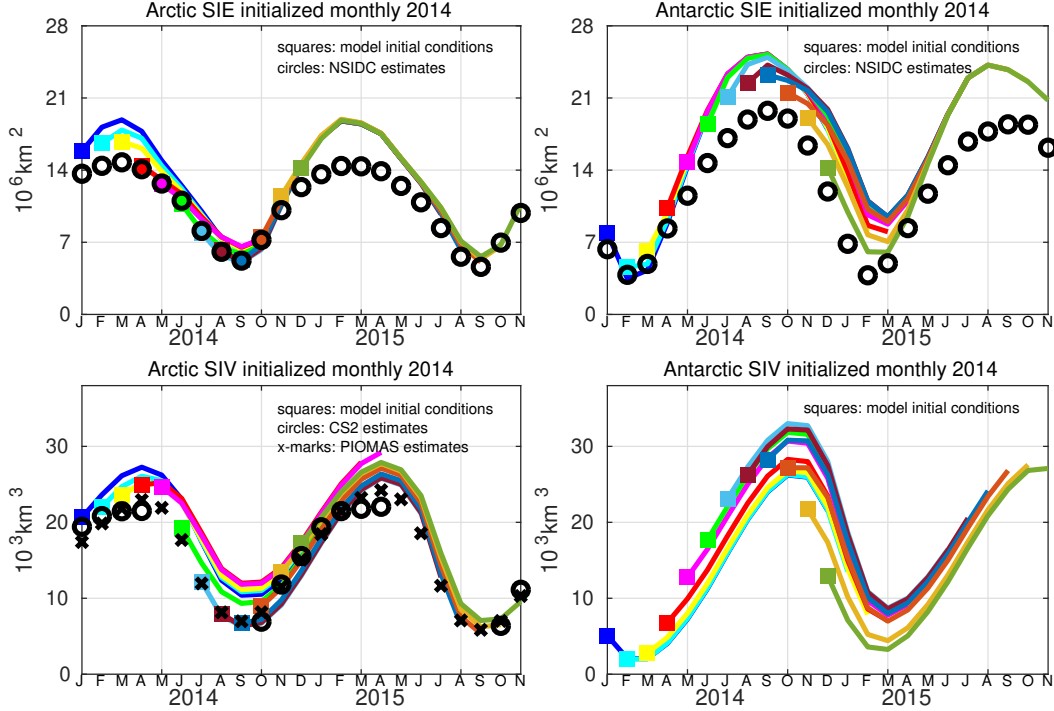

**Figure 1.** Sea ice extent (SIE, upper) and sea ice volume (SIV, lower) in the Arctic (left) and Antarctic (right) in the control experiments initialized monthly in 2014 and integrated for 12 months, with runs at different initialization times marked by different colors. Circles are observational estimates for SIE from NSIDC and SIV from CryoSat-2. SIV estimates from PIOMAS are marked by "x".

with James Rosinski, Ola Persson, Julie Schramm, Janet Intrieri, Chris Fairall, Antonietta Capotondi and Ligia Bernardet are also much appreciated. Benjamin W. Green helped with graphics. CFSR data used here are from the Research Data Archive, managed by the Computational and Information Systems Laboratory at the National Center for Atmospheric Research in Boulder, Colorado. The ice concentration data is obtained from the National Snow and Ice Data Center (NSIDC; http://nsidc.org/data/docs/daac/nsidc0051_gsfc_seaice.gd.html).



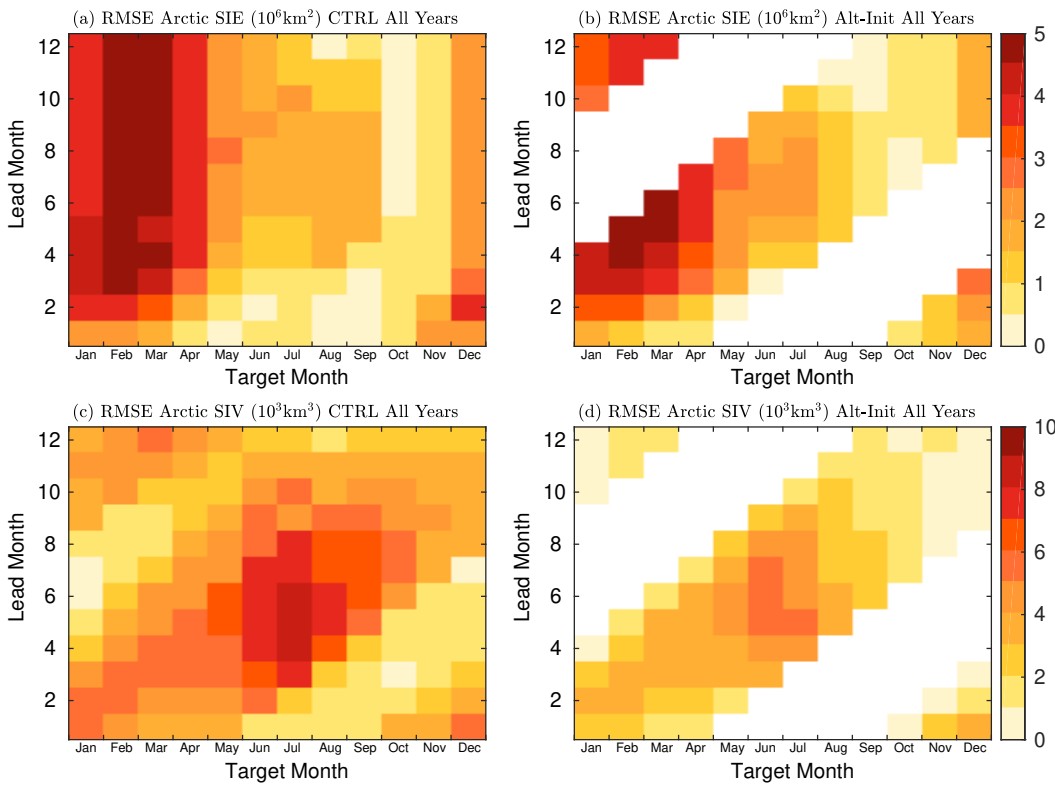

**Figure 2.** Root mean square error (RMSE) of the Arctic sea ice extent with respect to NSIDC (upper), and Arctic sea ice volume w.r.t. PIOMAS (lower), plotted at lead times up to 12 months against target months. Averaged over all years. Left: control experiments; Right: same as control, but initialized with CryoSat-2 Arctic sea ice thickness during October to April. White spaces indicate lack of CryoSat-2 data from May to September.

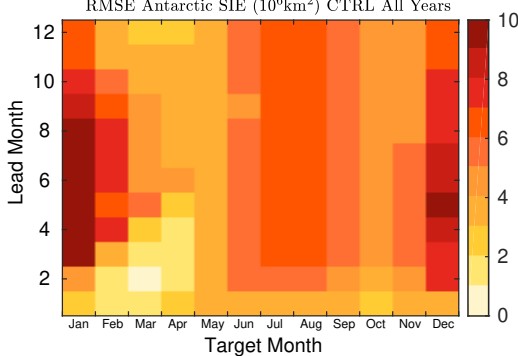

**Figure 3.** RMSE of the Antarctic sea ice extent w.r.t. NSIDC, shown at lead times up to 12 months against target months. Averaged over all years.

**Figure 4.** Upper row: Arctic sea ice extent used as initial conditions in the control (left) and alt-init (middle) experiments, and AMSR2 observations (right), all on April 1, 2016. Lower row: 6-month prediction of SIE for October 2016 for the control (left) and alt-init (middle) experiments, and AMSR2 observations in October 2016 (right).





**Figure 5.** Same as Fig. 4, except for sea ice volume and CryoSat-2 observations.



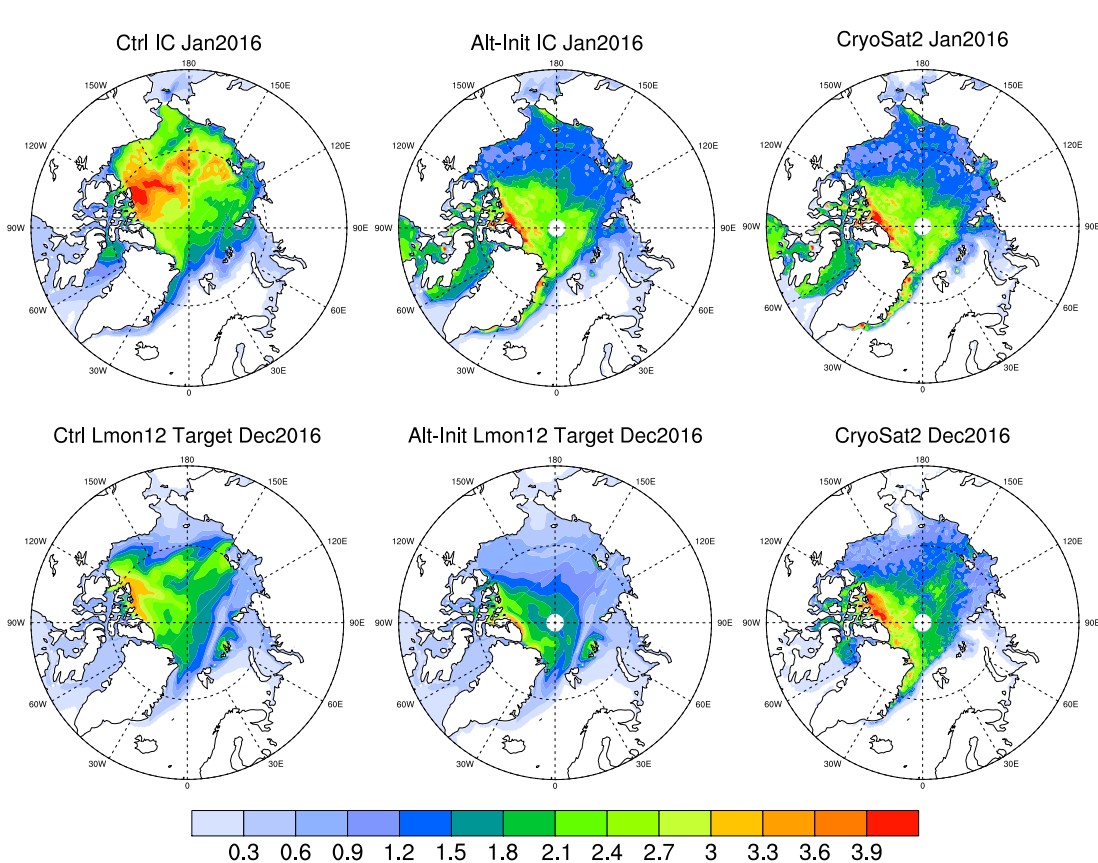

**Figure 6.** Same as Fig. 5, but initialized in January 2016 with a target month of December 2016.

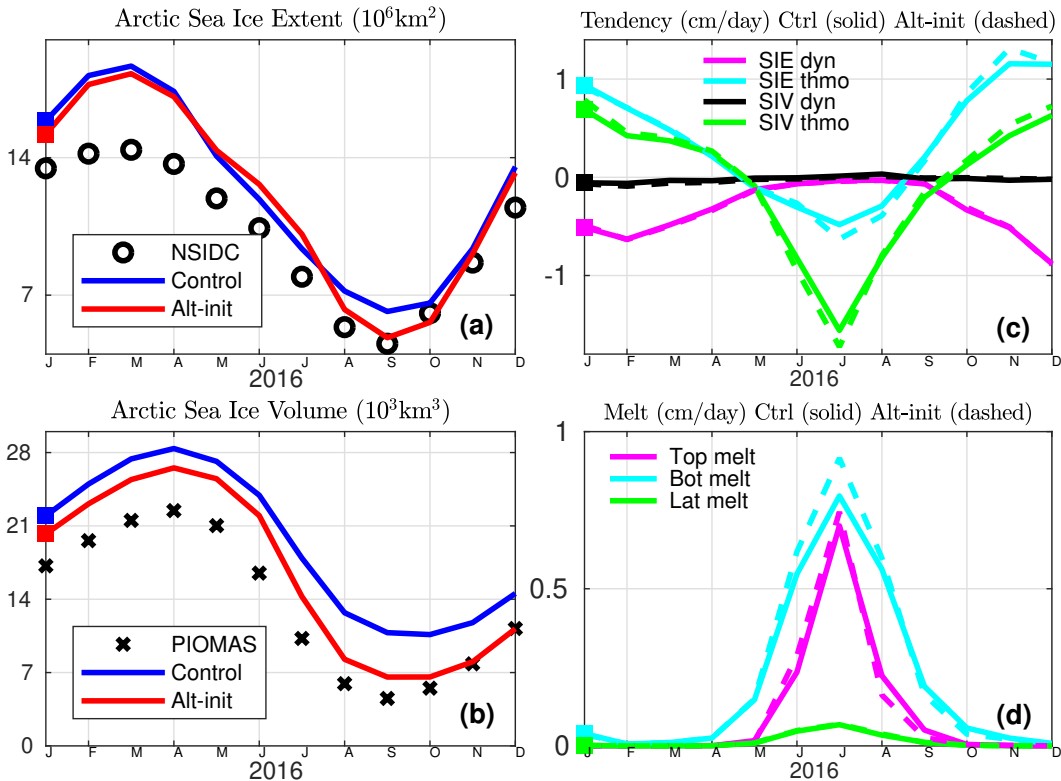

**Figure 7.** Arctic SIE and SIV in the control and alt-init experiments as well as estimates from NSIDC and PIOMAS are shown in (a) and (b). The mean tendency of SIE and SIV north of $67.5°N$ attributed to ice dynamics and thermodynamics is shown in (c). The mean top, bottom and lateral melt north of $67.5°N$ are shown in (d) for the control (solid lines) and alt-init (dashed lines). All from 12-month runs initialized in January 2016.

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





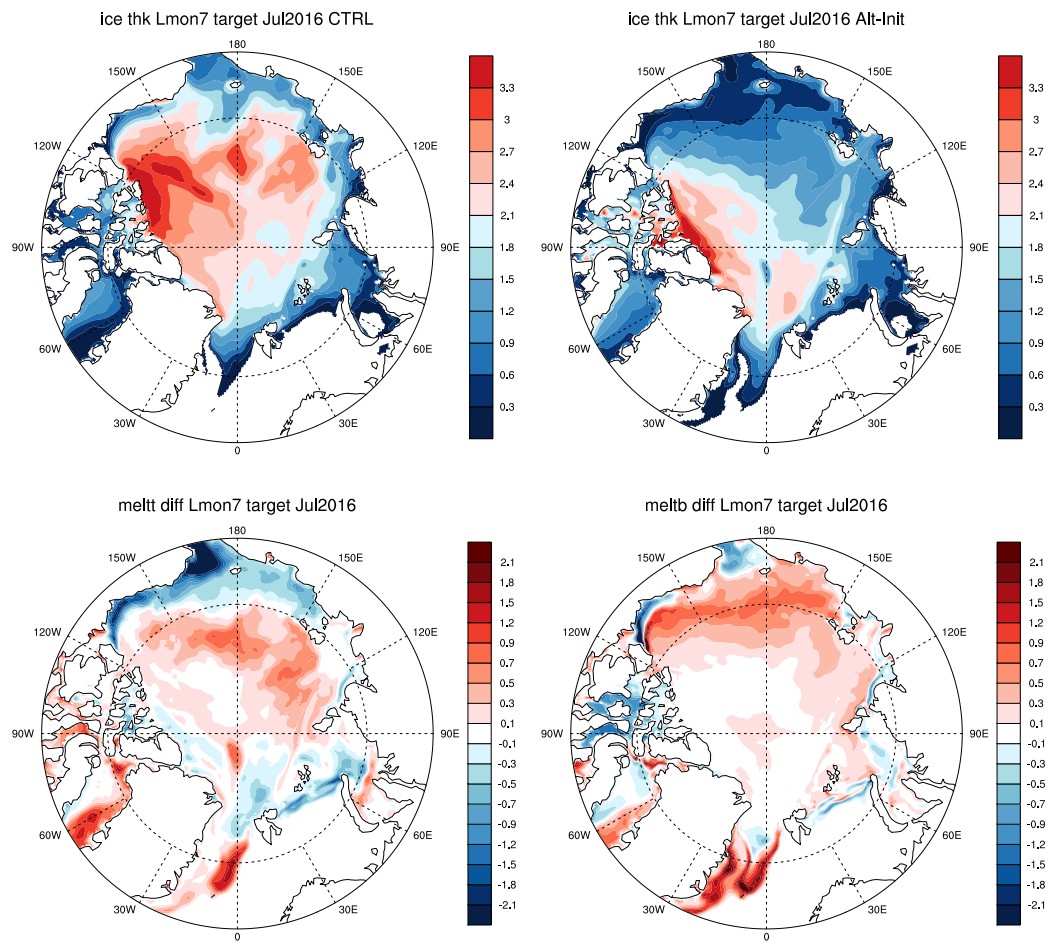

**Figure 8.** Upper panel: Arctic SIV in the control (left) and alt-init (right) experiments. Lower panel: differences in the top melt (left) and bottom melt (right) from the alt-init and control experiments. Initialized in January 2016 with a target month of July 2016.



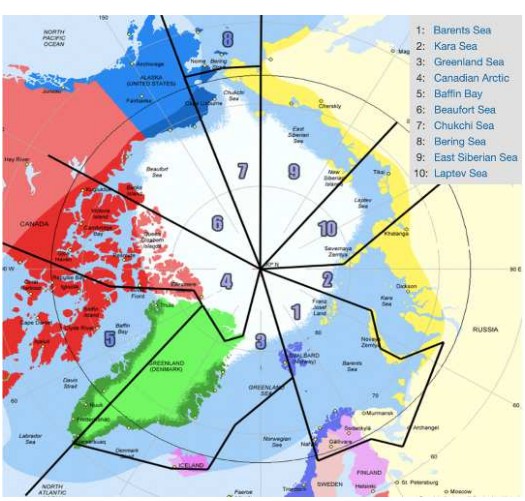

**Figure 9.** Different regions in the Arctic Ocean. Reproduced from https://arctic-roos.org.

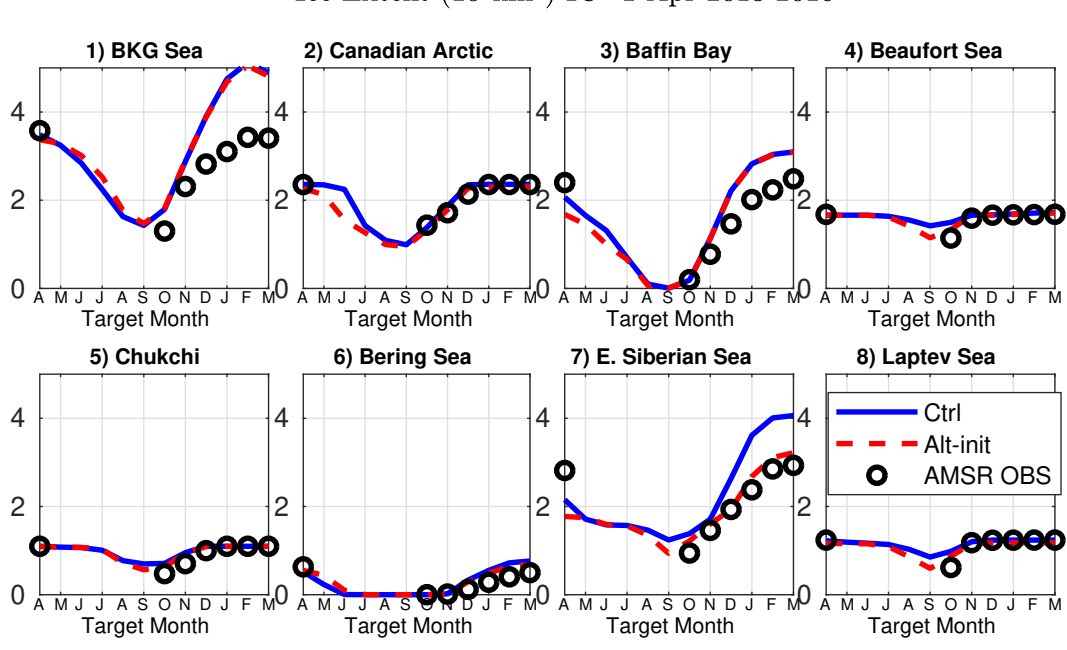

**Figure 10.** 12-month integration of SIE initialized in April in the control (solid blue) and alt-init runs (dashed red), averaged from 2013 to 2017, and AMSR2 observations (circles) in each basin where (1) represents Barents, Kara, and Greenland Seas (BKG) combined.





Ice Volume ($10^3$km$^3$) IC=1 Apr 2013-2016

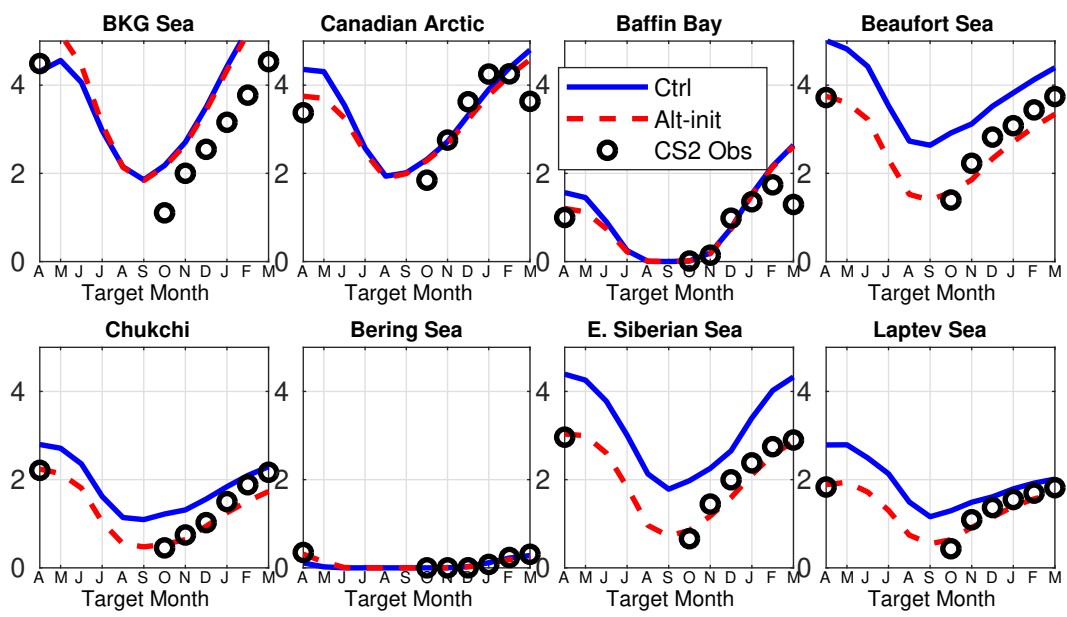

**Figure 11.** Same as Fig. 10, except for SIV, where circles mark CryoSat-2 observations.

Bleck, R. and Sun, S.: Diagnostics of the oceanic thermohaline circulation in a coupled climate model, Global and Planetary Change, 40, 233–248, 2004.

Blockley, E. W. and Peterson, K. A.: Improving Met Office seasonal predictions of Arctic sea ice using assimilation of CryoSat-2 thickness, The Cryosphere, 12, 3419–3438, 2018.

Bushuk, M., Msadek, R., Winton, M., Vecchi, G., Yang, X., Rosati1, A., and Gudgel, R.: Regional Arctic sea–ice prediction: potential versus
operational seasonal forecast skill, Climate Dynamics, 52, 2721–2743, 2019.

Bushuk, M., Winton, M., Bonan, D. B., Blanchard-Wrigglesworth, E., and Delworth, T. L.: A mechanism for the Arctic sea ice spring predictability barrier, Geophysical Research Letters, 47, 1–13, 2020.

Chapman, W. and Walsh, J.: Observed climate change in the Arctic, updated from Chapman and Walsh, 1993, Bulletin of the American Meteorological Society, 74, 33–47, 2003.

Chevallier, M. and Salas-Mélia, D.: The role of sea ice thickness distribution in the Arctic sea ice potential predictability: A diagnostic approach with a coupled GCM, Journal of Climate, 25, 3025–3038, 2012.

Day, J. J., Hawkins, E., and Tietsche, S.: Will Arctic sea ice thickness initialization improve seasonal forecast skill?, Geophysical Research Letters, 41, 7566–7575, 2014a.

Day, J. J., Tietsche, S., and Hawkins, E.: Pan-Arctic and regional sea ice predictability: Initialization month dependence, Journal of Climate,
27, 4371–4390, 2014b.



Dethloff, K., Rinke, A., Benkel, A., Køltzow, M., Sokolova, E., Saha, S. K., Handorf, D., Dorn, W., Rockel, B., von Storch, H., Haugen, J. E., Røed, L. P., Roeckner, E., Christensen, J. H., and Stendel, M.: A dynamical link between the Arctic and the global climate system, Geophysical Research Letters, 33, L03 703, 2006.

Grosfeld, K., Treffeisen, R., Asseng, J., Bartsch, A., Bräuer, B., Fritzsch, B., Gerdes, R., Hendricks, S., Hiller, W., Heygster, G.,
Krumpen, T., Lemke, P., Melsheimer, C., Nicolaus, M., Ricker, R., and Weigelt, M.: Online sea-ice knowledge and data platform <www.meereisportal.de>, Polarforschung, 85, 143–155, 2016.

Grumbine, R.: Long Range Sea Ice Drift Model Verification, MMAB Tech Note, 315, 22 pp, 2003.

Hebert, D. A., Allard, R. A., Metzger, E. J., Posey, P. G., Preller, R. H., Wallcraft, A. J., Phelps, M. W., and Smedstad, O. M.: Short-term sea ice forecasting: An assessment of ice concentration and ice drift forecasts using the U.S. Navy's Arctic Cap Nowcast/Forecast System, J.
Geophys. Res. Oceans, 120, 1–19, 2015.

Holland, M. M. and Bitz, C. M.: Polar amplification of climate change in coupled models, Climate Dynamics, 21, 221–232, 2003.

Holland, M. M., Bailey, D. A., and Vavrus, S.: Inherent sea ice predictability in the rapidly changing Arctic environment of the Community Climate System Model, version 3, Climate Dynamics, 36, 1239–1253, 2011.

Hunke, E. C., Lipscomb, W. H., Turner, A. K., Jeffery, N., and Elliott, S.: CICE: the Los Alamos sea ice model documentation and software
user's manual version 5.1, 2015.

Intrieri, J. M., Solomon, A., Cox, C., Persson, O., de Boer, G., Hughes, M., and Capotondi, A.: Sub-seasonal forecasting of the coupled Arctic system: Evaluation of the NOAA Experimental Coupled Arctic Forecast System (CAFS), manuscript submitted for publication, 2020.

IPCC: Climate Change 2014: Synthesis Report, Contribution of Working Groups I, II and III to the Fifth Assessment Report of the Intergov-
ernmental Panel on Climate Change [Core Writing Team, R.K. Pachauri and L.A. Meyer (eds.)]. IPCC, Geneva, Switzerland, 2014.

IPCC: Summary for Policymakers. In: Climate Change 2021: The Physical Science Basis, Contribution of Working Group I to the Sixth Assessment Report of the Intergovernmental Panel on Climate Change [Masson-Delmotte, V., P. Zhai, A. Pirani, S. L. Connors, C. Péan, S. Berger, N. Caud, Y. Chen, L. Goldfarb, M. I. Gomis, M. Huang, K. Leitzell, E. Lonnoy, J.B.R. Matthews, T. K. Maycock, T. Waterfield, O. Yelekçi, R. Yu and B. Zhou (eds.)]. Cambridge University Press. In Press, 2021.

Jaiser, R., Dethloff, K., Handorf, D., Rinke, A., and Cohen, J.: Impact of sea ice cover changes on the Northern Hemisphere atmospheric winter circulation, Tellus A: Dynamic Meteorology and Oceanography, 64, 11 595, 2012.

Krinner, G., Rinke, A., Dethloff, K., and Gorodetskaya, I.: Impact of prescribed Arctic sea ice thickness in simulations of the present and future climate, Climate Dynamics, 35, 619–633, 2010.

Li, X., Gerber, D. M. H. E. P., and Yoo, C.: Impacts of the north and tropical Atlantic Ocean on the Antarctic Peninsula and sea ice, Nature,
505, 538–542, 2014.

Liu, J., Song, M., Horton, R. M., and Hu, Y.: Revisiting the potential of melt pond fraction as a predictor for the seasonal Arctic sea ice extent minimum, Environ. Res. Lett., 10, 054 017, 2015.

Meier, W. N., Stroeve, J., Barrett, A., and Fetterer, F.: A simple approach to providing a more consistent arctic sea ice extent time series from the 1950s to present, Cryosphere, 6, 1359–1368, 2012.

Metzger, E. J., Smedstad, O. M., Thoppil, P. G., Hurlburt, H. E., Cummings, J. A., Wallcraft, A. J., Zamudio, L., Franklin, D. S., Posey, P. G., Phelps, M. W., Hogan, P. J., Bub, F. L., and Dehaan, C. J.: US Navy operational global ocean and Arctic ice prediction systems, Oceanography, 27, 32–43, 2014.





Palmer, T.: Predictability of weather and climate: From theory to practice, In T. Palmer and R. Hagedorn (Eds.), Predictability of Weather and Climate (pp. 1-29). Cambridge: Cambridge University Press, 2006.

Randall, D., Curry, J., Battisti, D., Flato, G., Grumbine, R., Hakkinen, S., Martinson, D., Preller, R., Walsh, J., , and Weatherly, J.: Status of and Outlook for Large-Scale Modeling of Atmosphere–Ice–Ocean Interactions in the Arctic, Bulletin of the American Meteorological Society, 79, 197–220, 1998.

Roberts, A., Craig, A., Maslowski, W., Osinski, R., Duvivier, A., Hughes, M., Nijssen, B., Cassano, J., and Brunke, M.: Simulating transient ice-ocean Ekman transport in the Regional Arctic System Model and Community Earth System Model. 211-228.
doi:10.3189/2015AoG69A760, Annals of Glaciology, 56, 211–228, 2015.

Saha, S., Moorthi, S., Pan, H.-L., Wu, X., Wang, J., Nadiga, S., Tripp, P., Kistler, R., Woollen, J., Behringer, D., Liu, H., Stokes, D., Grumbine, R., Gayno, G., Wang, J., Hou, Y.-T., ya Chuang, H., Juang, H.-M. H., Sela, J., Iredell, M., Treadon, R., Kleist, D., Delst, P. V., Keyser, D., Derber, J., Ek, M., Meng, J., Wei, H., Yang, R., Lord, S., van den Dool, H., Kumar, A., Wang, W., Long, C., Chelliah, M., Xue, Y., Huang, B., Schemm, J.-K., Ebisuzaki, W., Lin, R., Xie, P., Chen, M., Zhou, S., Higgins, W., Zou, C.-Z., Liu, Q., Chen, Y., Han, Y., Cucurull, L.,
Reynolds, R. W., Rutledge, G., and Goldberg, M.: NCEP Climate Forecast System Reanalysis, Bulletin of the American Meteorological Society, 91, 1015–1058, 2010.

Schröder, D., Feltham, D. L., Tsamados, M., Ridout, A., and Tilling, R.: New insight from CryoSat-2 sea ice thickness for sea ice modelling, The Cryosphere, 13, 125–139, 2019.

Schweiger, A., Lindsay, R., Zhang, J., Steele, M., and Stern, H.: Uncertainty in modeled arctic sea ice volume, J. Geophys. Res., 116,
C00D06, 2011.

Shaffrey, L. C., Stevens, I. T., Norton, W. A., Roberts, M. J., Vidale, P. L., Harle, J. D., Jrrar, A., Stevens, D. P., Woodage, M. J., Demory, M.-E., Donners, J., Clark, D. B., Clayton, A., Cole, J., Wilson, S. J., Connolley, W. M., Davies, T., Iwi, A., Johns, T. C., King, J. C., New, A. L., Slingo, J. M., Slingo, A., Steenman-Clark, L., and Martin, G. M.: U.K. HiGEM: The New U.K. High-Resolution Global Environment Model - Model Description and Basic Evaluation, Journal of Climate, 22, 1861–1896, 2009.

Wang, W., Chen, M., and Kumar, A.: Seasonal prediction of Arctic sea ice extent from a coupled dynamical forecast system, Mon. Weather Rev., 141, 1375–1394, 2013.