# Peer review of "Seasonal Sea Ice Prediction with the CICE Model and Positive Impact of CryoSat-2 Ice Thickness Initialization"

_The Cryosphere, 2021_

## Referee Comment (RC3)

[referee-annotated manuscript omitted]

---

## Author Comment (AC1)

We thank the reviewer for their careful reading of the manuscript and constructive remarks, which helped improve the quality of the manuscript. We have accepted all the recommendations, see details below (reviewer's comments in black, our replies in blue).

While investigating the origin of the line of low concentration north of Franz Josef Land in Fig.4, we found a bug in the namelist used to run CICE. We have rerun all the experiments with the corrected namelist. The revised results don't change the previous conclusions, but do alter figures at different magnitudes.

This manuscript analyzes impacts on sea ice thickness initialization on the simulation of sea ice extent and sea ice volume with the Los Alamos sea ice model (CICE) by comparing two sets of experiments initialized from the Climate Forecast System Reanalysis (CFSR) and CryoSat-2 satellite observations. The analysis of the experiments confirms results from earlier studies on initial sea ice thickness impacts on seasonal sea ice predictions. The manuscript is well structured and presentation of the results is reasonably clear. I would recommend acceptance of this manuscript for publication with a few minor revisions as listed below.

Lines 30-32: Suggest adding Collow's study to the citation. The work by Collow et al. (2015) is one of the earliest studies specifically on the need for improved sea ice thickness initial conditions. (Collow, T. W., W. Wang, A. Kumar, and J. Zhang, 2015: Improving Arctic sea ice prediction using PIOMAS initial sea ice thickness in a coupled ocean-atmosphere model. Mon. Wea. Rev., 143, 4618-4630. DOI: 10.1175/MWR-D-15-0097.1).

Done.

Line 41: Suggest indicating that the UFS is to be the next NOAA's operational coupled atmosphere-ocean-sea ice-land system for S2S predictions.

Done.

Fig. 1: Reduce thickness of the curves so the differences can be seen more clearly.

Done. The thicker lines were used initially for shown in ppt. Now with thinner lines, figures look more elegant. We have redone all line-figures with thinner lines in this manuscript. Thanks.

Lines 84-84; "The 12 once-per-month runs in 2014 are shown here as an example, as year to year variations are relatively small". Does this mean the amplitude of interannual variations is smaller than that of model errors?

We meant to say the conclusion from year 2014 in Fig.1 is valid for all other years, despite year to year variations. We redid Fig.1 with an all-year mean and standard deviation. This sentence is rewritten as:

"The Arctic SIE forecast matches the NSIDC observations better in the warm season than in the cold season both at 0.5-month and 5.5-month lead times, and the positive bias is biggest in

winter. The biggest interannual variabilities in the predicted Arctic SIE during this period are in September as shown by the standard deviation."

Lines 87-89: "The Arctic SIE forecast matches observations better in the warm season than in the cold season at all lead times, and a positive SIE bias is seen in the cold season. When the SIV in the Arctic is higher than observations or reanalysis to begin with, this positive SIV bias often remains in the model throughout the forecast.". Although initial sea ice thickness may have some impacts in the cold season, its impact is more significant during the melt season because it directly affects the melt rate. The larger SIE error in the cold season could be related to other factors such as model physics, atmospheric forcing, initial ocean state, and ocean dynamical processes.

This is exactly the case. We have replaced Fig.1 with a multi-year mean and standard deviation and modified this paragraph in the manuscript to:

"The Arctic SIV forecast at 0.5-month lead is higher than CryoSat-2 observations or PIOMAS reanalysis in the cold season. At 5.5-month lead time, the biggest positive bias in SIV is shifted to the warm season, suggesting the bias might be stemmed from the cold season. The impact of the positive SIV bias appears to be more significant during the melt season as the ice thickness directly affects the basal melting rate as shown later in Figs. 7 and 8."

Lines 94-97 and Fig 2: To see the comparison between CTRL and Alt-Init more clearly, I suggest adding two panels to show the differences between CTRL and Alt-Init, one for SIE and the other for SIV. For SIE, it looks like the improvement in the summer melt season (Jun-Sep) is larger due to the use of better initial sea ice thickness.

Thanks for this suggestion. We have added a difference plot in Fig. 2, which shows the buildup of bias in ice thickness with time much better.

Fig. 7: I suggest making the curves 7c and 7d thinner.

Done.

Lines 140-141: "Apparently, there are more bottom and top melt in the Beaufort, Chukchi and East Siberian Seas in the alt-init run than in the control run, …". It looks to me the Alt-Init produces less top melt (blue colors) in Chukchi Sea and East Siberian Sea, and a large part of Beaufort Sea in the lower-left panel of Fig. 8.

Sorry for the oversight. We left out the keyword "poleward". Now the modified text has changed to:

"Compared to the control experiment, the biggest difference is a larger basal melt in the Arctic region between 120°E and 120°W in the alt-init experiments, where the ice thickness is mostly below 1.5 m compared to above 1.5 m in the control experiment."

---

## Author Comment (AC2)

Review of "Seasonal Sea Ice Prediction with the CICE Model and Positive Impact of CryoSat-2 Ice Thickness Initialization" by Shan Sun and Amy Solomon.

This manuscript investigates the forecast skill of the sea-ice model CICE in stand-alone simulations. The analysis includes both the Arctic and Antarctic. The model CICE is shown to adequately reproduce the seasonality of the sea ice extent and volume, but with a larger difficulty in forecasting the winter and spring conditions. A particular interest is placed on the influence of the initial ice thickness on the forecast skill. Specifically, the use of the CFSR ice thicknesses, which over-estimates the ice thickness, is shown to decrease the forecast skills regardless of the initiation month. This is largely improved when ice thickness from Cryo-Sat2 is used instead. The simulations are presented as a baseline for future studies on the forecast skill of coupled models using CICE as the sea-ice component.

This manuscript is relatively well written, although many sentences are too long and confusingly constructed. The results are interesting and will interest many in sea-ice modelling community. Nonetheless, the manuscript suffers from an unclear problem statement in the introduction and from a tendency to describe figures without much in depth interpretations. This is especially important given that other studies have looked into the impact of ice thickness on forecast skills.

I believe this manuscript have potential for publication, but require major revisions.

Major points:

There is a tendency (mostly in the introduction) in lumping too many ideas in complicated sentences. The text should be revised in that regard.
The introduction lacks a problem statement and should be revised to clearly identify the scientific questions that the analysis aims to answer. A problem statement is vaguely formulated at L37-45 but mixed with the broader context. I believe that adding a paragraph devoted to the problem statement (mainly on the influence of the initial ice thickness on the forecast skills) would largely clarify the scope of the manuscript. In particular, it should clarify what information the stand-alone CICE simulations can bring that has not yet been documented.
There is very little mention of the sea-ice dynamics (in the experiment setup, results and discussion) although it should largely affect the sea-ice extent, especially in a year-long simulation. The thermodynamics and dynamics contribution to the SIE is briefly investigated in Figure 7, although this analysis not clear (the methods are not described) and needs to be clarified.
Much of the results are very descriptive and not thoroughly discussed. I believe that a comprehensive assessment needs to be include before publication. For instance, many statements are vague and general (e.g. the forecast skill is reduced in the alt-init simulations), despite the figures presenting much information. More in depth analysis of the results could include, for instance, explaining the differences in the SIC and SIV forecast skill patterns, and how it relates to the initial ice thickness.

We thank the reviewer for their careful reading of the manuscript and many constructive suggestions, which helped improve the quality of the manuscript. We have incorporated them in the revision, see details below (reviewer's comments in black, our replies in blue).

 Specific points:

L17-19: This sentence is confusingly constructed. Perhaps dividing it in smaller sentences would be clearer.

We have simplified this sentence to:

"A relatively thin material layer between the atmosphere and ocean, sea ice amplifies radiative climate feedback with a higher surface albedo than open water."

L20-22: This sentence is currently confusing as it covers too much while being too vague. For instance, what "forecast" and "predictability" are we talking about (weather conditions? Ocean? Climate?). The vague reference to the impact of sea ice conditions on teleconnections also needs expanding.

This sentence is rewritten as:

"Reliable sea ice prediction is important not only for the polar regions but also is expected to improve predictability at mid-latitudes at subseasonal to seasonal time scales due to teleconnections."

L23: It is not clear what we are talking about here. Weather?

This sentence is rewritten as:

"Finding sources of weather predictability at S2S time scales is a challenging research topic."

L23: I am not sure that "leading" is the right verb... Perhaps "Driving"?

This sentence is rewritten as

"Finding sources of predictability for weather at S2S time scales is a challenging research topic."

L29: The use of "In particular" is confusing here, as we were discussing the influence of sea ice on weather predictability, but now jump to sea ice forecasting.

This sentence is rewritten as:

"Weather predictions from a numerical model incorporating a sea ice model have higher skill than those based on persistence of Arctic sea ice (Grumbine, 2003; Hebert et al., 2015; Intrieri et al., 2022). In addition, sea ice forecasts at seasonal to interannual timescales relies on the accuracy of the sea ice initial conditions (Holland et al., 2011; Blanchard-Wrigglesworth et al.,

2011b; Wang et al., 2013) and in particular, a more realistic sea ice thickness initialization (Krinner et al., 2010; Chevallier and Salas-Mélia, 2012; Day et al., 2014a; Collow et al., 2015; Allard et al., 2018; Blockley and Peterson, 2018; Schröder et al., 2019, to name a few)."

L37-39: Do I understand that here, you validate the sea ice model component of a fully coupled ice-ocean-atm model, in a first step towards investigating how it feedbacks with the other components?

The standalone sea ice model is evaluated here, while a fully coupled ice-ocean-atm model is under development. We have modified this paragraph to:

"In order to separate various feedbacks among the components of a fully coupled model, in this study we aim to evaluate the seasonal prediction skill of CICE in standalone mode with prescribed atmospheric forcings. It is essential to assess the skills of each module separately in an uncoupled mode before assembling them in a fully coupled model, as it is easier to reveal the strength and weakness of a module in such a controlled environment. This work is even more relevant as CICE, originally developed for climate research, is to be used in the S2S prediction in operation."

L37-45: The structure of this paragraph makes it difficult to understand the scope of the paper. It first indicates that the aim is to isolate feedback processes between coupled model components, then that it is to validate the sea-ice model used in NOAA UFS, in stand-alone simulations. However, I believe that the real goal here is to assess the influence of initial thicknesses on the ice predictability within CICE. This needs to be clarified.

The initial goal of the project was to evaluate the model skill with a given set of initial conditions and atmospheric forcing. Half way through the project, we noticed an obvious bias in the ice thickness in the 'given' initial conditions. We decided to reduce this bias by using satellite observations and evaluate how much skill can be gained from this alone. We have made this clearer in the text.

"We also investigate the sensitivity of prediction skill in the standalone CICE when a bias in the ice thickness initialization is removed."

L51-53: This sentence needs revisions.

This sentence is rewritten as:

"The linear function of salinity (ktherm=1) was used for the freezing temperature. The elastic-viscous-plastic rheology (kdyn=1) were specified for the sea ice dynamics. The ice strength was set to be closely related to the ridging scheme (kstrength=1). In addition, CICE needs the information of sea surface temperature (SST) which can be either prescribed or generated by its own built-in mixed layer model. We chose the latter in this study for the sake of consistency between the SST and the ice state."

L55: I suggest starting a new sentence after "experiment".

Done.

Section 2: Some information on the dynamical component should be provided (e.g., I assume it is the standard EVP rheology and strength parameters in CICE ?).

Done. See above under L51-53.

L66-69: This long sentence could be improved.

This sentence is rewritten as:

"Furthermore, monthly mean estimates of the Arctic ice thickness are available from satellite observations of CryoSat-2 during boreal winter (October to April) since 2011 (Grosfeld et al., 2016). Additional CICE experiments were carried out by initializing ice thickness from this dataset instead of CFSR."

Section 3: What is defined as a "reliable forecast"? This statement is made at various places throughout the result section, but sounds rather subjective.

This sentence is rewritten as:

"The model is able to make SIE forecasts in good agreement with observations at all lead times for the austral winter, spring and especially fall."

L82: It should be specified here (not later) that Figure 1 only shows simulations from 2014.

We have modified Fig.1 with all-year mean and standard deviation to show the interannual variability.

L84: There is very significant inter-annual variability in Arctic (and Antarctic) sea ice extent, yet here you say that the inter-annual differences are small? This needs to be clarified. For instance, you show 2014, a year where the winter maximum was relatively small (~14.9 million km2) and the summer minimum remarkably large (5.0 million km2). It is possible that conclusions drawns from Figure 1 are not representative of different years, such as 2012.

We meant to say the conclusion from year 2014 in Fig.1 is valid for all other years, despite year to year variations. We redid Fig.1 with an all-year mean and standard deviation. This sentence is rewritten as:

"The Arctic SIE forecast matches the NSIDC observations better in the warm season than in the cold season both at 0.5-month and 5.5-month lead times, and the positive bias is biggest in winter. The biggest interannual variabilities in the predicted Arctic SIE occurs in September as shown by the standard deviation."

L94: This result is very surprising to me, as the summer minimum is usually described as being more difficult to forecast and more dependent on the early summer meteorological conditions.

This is indeed the case in many models. Here we see a spring barrier with a lower skill for the summer prediction in Fig.2a. We have added this sentence in the manuscript:

"The control experiments have the lowest RMSE in the Arctic SIE in fall at all lead times up to 12 months. There is an obvious skill barrier in spring for the summer forecast."

L91-102: Together, these results are confusing and should come with some analysis and explanation. The ice extent results seem to indicate that the model overestimates the ice growth, but that does not show in the SIV results. On the contrary, the SIV results seem to indicate biases in the spring and summer melt, but this does not show in the ice extent results. Why? Is this expected? Also, changing the initial thickness improves the overall skill but does not seem to change the temporal patterns. Does this implies that the thickness influences the error magnitude but not the predictability patterns?

We redid Fig.1 to show the biases in SIE and SIV in the control experiment at 0.5-month and 5.5-month lead time to show
(1) there is a positive bias in both SIE and SIV at 0.5-month lead time in the Arctic during the cold season;
(2) the delay of max/min peak in SIV with longer lead time indicates the positive bias in SIV has a long-lasting effect on SIV and SIE.

Then in Fig.2, we plotted the bias in SIE and SIV at all lead times in both control and alt-init experiments to show
(1) biases in SIE and SIV are in general smaller in the magnitude in the alt-init compared to the control experiment, while the bias pattern is still the same;
(2) the difference in SIV from "alt-init" and "control" is bigger with longer lead times in summer and fall forecast;
(3) the phase shift between the biggest bias in SIE and SIV is obvious.

L108: spatial distribution

Figure 2: A couple things that are concerning in these figures:

Why is there a low concentration spots at the North Pole, but only for the top-middle and bottom-right panels, and of different size?

We didn't realize that there was no measurement at the North Pole in the original CryoSat2 data set. We have filled up this hole.

What is that line of low concentration North of Franz Josef Land, running from the Laptev Sea to Svalbard? It is a very strange location and orientation for such an LKF, and it is seen both in April and October. More confusing, it is seen in the observations but only in October. Is it real or is it an artefact?

We investigated the origin of the line of low concentration north of Franz Josef Land, and found a bug in the namelist used to run CICE, which produced wrong wind and a divergence along the model grid line of i=1 where a bipolar grid was used. We have rerun all the experiments and this line is gone.

The AMSR2 observation data used in Fig. 4 was taken from day 1 of an additional model run initialized with AMSR2 (not documented here). However, the day 1 output in October was already impaired by the bug mentioned above when the ice was very thin. We have now replaced the figure by the "raw" AMSR2 data IC and the line is gone.

L117: On the initial thickness being the dominant source of error: how did you determine this dominance? There are no results on the contribution from other sources (dynamics mass balance, thermodynamics mass balances). This statement is also contradictory with the fact that the errors are large in the winter sea ice at all lead time: they are thus are likely dominated by other factors than initial ice thickness. The use of CryoSat also does not seem to change this pattern.

We verified the ice thickness against the CryoSat2 data set. We agree that there are other sources of errors since this model still has biases when initializing with CryoSat-2, as shown in Fig.2. We see the initial ice thickness is an "obvious" error at time=0. We have added a discussion in Sec 3.1. This section is rewritten as

"As seen in Figs. 4 and 5, a positive bias in SIT in the initial conditions appears to be one source of the error in SIT as well as SIA in the seasonal forecast. Both SIT and SIA forecasts improved when this bias was removed in the alt-init experiment."

L122-123: This sentence is confusing and needs to be re-organised.

This sentence is rewritten as

"Fig.6 presents SIV initially in January 2016, its forecast for December 2016 in both experiments and the comparison to CryoSat-2 observations."

L129-130: I would edit to: "the SIE and SIV in the alt-init run in the fall ended up closer to NSIDC and PIOMAS than in the control run."

Done.

L130-137: How are the dynamics and thermodynamics contribution measured? I find it somewhat surprising that the dynamics tendency is exclusively negative. Also unclear to me is how do you define the thermodynamics tendency in sea ice extent? The thermodynamics is usually defined by column physics, and not directly related to changes in area.

We made an error in the unit in Fig.7(c): the tendencies are measured in "%/day" for ice area and "cm/day" for ice volume per unit area. The area tendency from dynamics and thermo-dynamics (daidtd and daidtt) and the volume tendency from dynamics and thermo-dynamics (dvidtd and

dvidtt) are achieved in the standard model output, which are shown in the attached Figs. s1 and s2, valid for January 2016 and July 2016, with IC of January 1, 2016. The dynamics tendency for sea ice area (SIA) in January is a mixture of positive tendency near the ice edge and negative near coastlines, overall dominated by the negative. The dynamics tendency for SIA in July is smaller than its thermo-dynamic component.

The physical mechanism of sea ice dynamics can also be regarded as creating leads. Therefore, its tendency on SIA tends to be negative in all seasons.

L138-144: How can the top melt be influenced by the ice thickness? It is more intuitive for the bottom melt, as the reduced thickness would also reduce the insulation, but it could also be mentioned.

Unlike the basal melt, the top melt is not directly related to the ice thickness. Some surface properties like long wave emissivity and aerodynamic roughness are a function of the ice thickness. We have revised the text to…

"The biggest difference is a larger basal melt in the Arctic region between 120°E and 120°W in the alt-init experiments, where the ice thickness is mostly below 1.5 m compared to above 1.5 m in the control experiment. The basal melt pattern is consistent with the fact that the conductive heat flux at the ice bottom is inversely proportional to the ice thickness (Hunke, et al. 2015)."

L140: remove "apparently".

Done.

[Figure]

Fig. s1: top, from left to right: ice area, its dynamics and thermodynamics tendency, all monthly mean for January 2016; Bottom: same as top, except for the ice thickness. Initialized on January 1, 2016.

[Figure]

Fig.s2: same as in Fig.s1, except valid for July 2016.

---

## Author Comment (AC3)

Seasonal Sea Ice Prediction with the CICE Model and Positive Impact of CryoSat-2 Ice Thickness Initialization
Shan Sun and Amy Solomon

In this paper the authors analyse the seasonal prediction skill of a stand-alone CICE model forced/initialised using CFSR. The study is bipolar, although there is a much stronger focus on the Arctic. The authors show that initialising the model with observed sea ice thickness inferred from CryoSat-2 radar altimetry considerably improves the forecast skill, as has been shown previously for other models in other studies (as they correctly point out).

The manuscript is relatively well written and presented and the results will be of interest to the community. Therefore I think it worthy of publication in The Cryosphere.

However, the figures could do with a bit more attention in relation to figure captions and colourmaps. Furthermore, the study could be better motivated, and the discussion of the figures/results is often rather on the shallow side. I therefore recommend that this manuscript requires considerable revision before it is accepted for publication here.

We thank the reviewer for their careful reading of the manuscript and many constructive suggestions, including a detailed edit in the pdf file, which helped improve the quality of the manuscript. We have incorporated them in the revision, see details below (reviewer's comments in black, our replies in blue).

While investigating the origin of the low concentration line north of Franz Josef Land in Fig.4, we found a bug in the namelist used to run CICE. We have rerun all the experiments with the corrected namelist. The revised results don't change the previous conclusions, but do alter figures at different magnitudes.

**Particular points**

A detailed list of comments can be found in the attached pdf document but I highlight here a few points that will particularly need addressing.

the study needs to be a bit better motivated. The main motivation I can see for the study is lines 36-42 which states that fully coupled (AOIL) models are "considered the ultimate tool" (which incidentally would be considered an insult here!) for sea ice seasonal prediction but here the stand-alone model is used in order to "separate various feedbacks among the components of a fully coupled model". However, this separation of feedbacks is not done in the ensuing manuscript! It is also not mentioned anywhere (albeit a trivial point) that the stand-alone approach is much cheaper.

We have made this clearer in the revised manuscript:

"In order to separate various feedbacks among the components of a fully coupled model, in this study we aim to evaluate the seasonal prediction skill of CICE in a standalone mode with prescribed atmospheric forcings. It is essential to assess the skills of each module separately in

uncoupled mode before assembling them in a fully coupled model, as it is easier to reveal the strength and weakness of a module in such a controlled environment. This work is even more relevant as CICE, originally developed for climate research, is to be used in the S2S prediction in operation."

there is no consideration of internal variability, which is a huge factor for sea ice and in polar regions generally, or significance. Many of the figures contain means of multiple years of model runs, which could also include error bars or shading to help understand the impact of internal variability (or at least inter-annual variability over the study period). Likewise hatching could be added to difference plots to try and portray to the reader how significant the changes are in relation to natural/chaotic differences.

We have redone Fig.1 with an all-year mean and standard deviation to show the inter-annual variability over the 7 years. Now it is rewritten as:

"The Arctic SIE forecast matches the NSIDC observations better in the warm season than in the cold season both at 0.5-month and 5.5-month lead times, and the positive bias is biggest in winter. The biggest interannual variabilities in the predicted Arctic SIE occurred in September as shown by the standard deviation."

the CryoSat-2 data, and the way that it is used to initialise the model, are poorly described and so I am left wondering whether things have been done sensibly. There is no mention of what happens with thinner ice (for which CS-2 errors are near-infinite!) and no mention of what is done with the snow on top of the sea ice. Furthermore, it looks like they have not been very careful with their QC because the CryoSat-2 "pole hole" appears as open water in the sea ice concentration for their "alt-init" runs!

We didn't realize these issues in the CryoSat-2 dataset and thus didn't perform QC. The snow on top of the sea ice is prescribed as a part of the atmospheric forcings. We have filled the "pole hole" in the CryoSat-2 and added the following to the manuscript:

"The ice thickness from the CryoSat-2 is treated as one ice thickness category at each model grid in the CICE initialization. For grid cells starting with zero ice thickness in the CryoSat-2 data, the sea ice area from CFSR is reset to zero initially to be consistent."

>95% of the article is focussed on the Arctic but with approx. 4 sentences and a 1-panel figure on the Antarctic, which feels a bit orphaned within the bigger picture of this manuscript. I think the authors should drop the Antarctic and limit the scope of this study to focus on the Arctic only - particularly given that the impact of SIT initialisation cannot be evaluated there, which is actually the second half of the manuscript title!

The goal of the project was to evaluate sea ice forecasts both in the Arctic and Antarctic. We have added more discussion on Antarctica in Fig.1, in addition to Fig.3. We think there are some values in the discussion on Antarctica although we wish we could have more data to validate forecasts in the Antarctic.

the results are often only described in a very shallow way without any mechanisms or processes being given. For example, the increased basal & top melting for the runs with thinner sea ice is not obvious and so the mechanisms should be talked about

there is general confusion between 1D and 2D sea ice variables/quantities in the figures and accompanying text. For example, sea ice "extent", "area" and "concentration" seem to be used interchangeably and so are "thickness" and "volume"

many of the titles, legends and colourmaps used in the figures are not intuitive for the reader. there are also some 'rainbow' colourmaps, which are also problematic for people who suffer from colour-blindness.

(1) We have added discussion on increased basal melt:
"The biggest difference is a larger basal melt in the Arctic region between 120˚E and 120˚W in the alt-init experiments, where the ice thickness is mostly below 1.5 m compared to above 1.5 m in the control experiment. The basal melt pattern is consistent with the fact that the conductive heat flux at the ice bottom is inversely proportional to the ice thickness (Hunke, et al. 2015)."

(2) We fixed an error in the unit in Fig.7(c): the tendencies are measured in "%/day" for ice area and "cm/day" for ice thickness.

(3) We now distinguish 1D and 2D sea ice variables according to their unit throughout the manuscript:
- sea ice extent and volume are 1D variables in the unit of $m^2$ and $m^3$
- sea ice area and thickness are 2D variables in the unit of % and m

(4) We have replaced the rainbow colorbar in Figs. 4, 5, 6 and 8 by color-blindness friendly colorbars as much as possible.

**Comments in Supplement**:

Thanks for your feedback on the pdf file. It is greatly appreciated. We have addressed all your comments - some may be redundant with above.

Line 7: added "this bias"
Line 8: Is this true? The rate at which sea ice melts altogether is lower of course (as there's more ice) but is it true that surface and basal melting rates are impacted? Obviously basal growth rate is hugely dependent on thickness but how is melting? Can you describe the mechanisms?

Unlike the basal melt, the top melt is not directly related to the ice thickness. Some surface properties like long wave emissivity and aerodynamic roughness are a function of the ice thickness. We have revised the text in the abstract as follow:

"In addition, thicker ice has a lower basal melting rate in the warm season, contributing to this positive bias."

and added discussion on mechanisms in Fig.8:

"The biggest difference is a larger basal melt in the Arctic region between 120°E and 120°W in the alt-init experiments, where the ice thickness is mostly below 1.5 m compared to above 1.5 m in the control experiment. The basal melt pattern is consistent with the fact that the conductive heat flux at the ice bottom is inversely proportional to the ice thickness (Hunke, et al. 2015)."

Line 10: Changed "indicates" to "confirms"
Line 14: Removed discussion on ice area reduction in Antarctica.
Line 21: Changed to "in mid-latitudes"
Line 28: Changed to "predictive skill"
Line 31: Changed to "sea ice volume has more persistence than sea ice area"
Line 40: We are now referring to Hunke et al. 2020, as it included more information than in Tonani et al. 2015, and removed Shaffrey et al. (2009).
Line 50: We use ktherm=1, and have removed "mushy physics".
Line 68: Do you have a paper reference for these observations. I'm surprised that in Figure 1 your CS2 volumes are lower than for the PIOMAS model when many other studies have shown the opposite. For example Figure 3 of Laxon et al. (2013) [https://agupubs.onlinelibrary.wiley.com/doi/full/10.1002/grl.50193] and Figure 2 of Tilling et al., (2015) [https://www.nature.com/articles/ngeo2489].
The latter of these papers includes some overlap with the period that you are running here and the CS2 estimates look much closer to your model!

Are the AWI CS2 data really that different from the CPOM data? Most papers I've seen have said they are 'similar'!

We have replaced Fig.1 using the all-year mean and the standard deviation, and recalculated the Arctic SIV values from the CryoSat-2 data on its native grid instead of on the CICE model grid. That is why the Arctic SIV from the CryoSat-2 here is lower than in the original version.

As for the lower-left panel of Fig. 1,
   (1) The CryoSat-2 values in Fig.1 are consistent with Laxon et al. (2013)
   (2) The PIOMAS values in Fig.1 are consistent with what is shown at the PIOMAS website. It is unclear why the PIOMAS values in Laxon et al. (2013) are lower than what is shown at the PIOMAS website.

We compared the SIV from CryoSat-2 and PIOMAS during 2011-2017, see Fig.s3 attached, where PIOMAS is mostly higher than CryoSat-2 in winter and spring. The two datasets are comparable in fall.

Line 69: What do you do with the snow cover? Does that just stay as it is in CFSR?
The snow is arguably more important than the ice itself - especially for the evolution of the ice pack through winter (snow conductivity ~10x ice conductivity) but also for the onset of melting (higher albedo).

The snow cover is prescribed using CFSR as a part of the atmospheric forcings. The snow melting rate is added to Fig. 7, where the snow melt peaks one month earlier than the melt from top, basal and lateral.

Also how do you account for the fact that CS2 are not valid for thin ice (errors tangent to infinity as shown by Ricker et al., 2017, https://tc.copernicus.org/articles/11/1607/2017/).
Is the CS2 data you use QC'd with thin values removed or do you use them despite the high errors? If the former then how do you ensure not to set the SIC to zero too?
It might have been better to have used the blended CS2+SMOS product of Ricker et al.?

We were unaware of the QC issues with the CS2 data for thin ice and therefore used it as is. Sea ice area is set to zero at all points with zero ice thickness initially to be consistent. We will certainly use the blended CS2+SMOS product of Ricker et al. in the future application.

Line 71: It looks like you might have a problem with this approach because it doesn't seem to differentiate between areas where the CS2 thickness is not defined and where it is actually zero. By this I mean that you have a "pole hole" in Fig 4 (b) that is not in 4 (a) and so much be caused by the mechanism.
So as well as starting with thinner ice you will also have lower initial extent and more local ocean heating through the summer?

(1) We have filled up the missing value at the "pole hole" in the CS2 data.
(2) For other missing data in the CryoSat-2 dataset, we treated ice thickness to be zero, and then set the ice area to zero at all points with zero ice thickness.
(3) The errors in CS2 could contribute to the model bias as well. We have added this to the manuscript.

Table 1, This confused me a bit because the table is the transpose of what I'd expect.
Done. Now we managed to fit 4 columns across the width of the page.

Line 91: How is this calculated? Are you calculating the basin-scale extent (SIE) for each dataset and then calculating the RMSE of the yearly values for each month? Or are you accounting for spatial variability (similar to IIEE)?

Integrated ice-edge error (IIEE) is a very informative metric. In this study, we used the basin-scale SIE and SIV values in the standard CICE model output, as we find these values represent the messages we try to deliver here. As shown in Fig.s4, since both model experiments tend to overestimate the ice edge along the Arctic periphery, there is either a positive or near zero bias at each longitude. Thus the basin-scale SIE is a valid metric to use in this application.

Line 95: I would expect this to be driven by biases in the ocean given that the winter ice edge is controlled by the SSTs (i.e., how far south the ice can get).
One of the primary mechanisms for Arctic winter sea ice melting is northward heat transport into the GIN & Barents Seas.

Do you think this is likely caused by the fact you run the model stand-alone with only a mixed-layer ocean (i.e., no horizontal transport)?

If so (or anyway!) it would be good to include mention of this.

We agree and have added this:

"It is most likely due to the lack of interaction with the Atlantic Ocean, the mechanism that is missing in this experiment by using a simplified mixed layer ocean model mentioned earlier. Without northward oceanic heat transport from the Atlantic, the sea ice edge tends to extend too far southward in winter."

Line 103: Lines 103-106 and Figure 3 represent the entirety of the Antarctic sea ice evaluation in this study. Much of what is said is that the observational coverage is too poor to do more (SIV anyhow).
So my obvious question is whether this brings any value to the paper?

My thoughts are that it might be better to drop these 4 lines and Figure 3 and focus the paper on the Arctic. Particularly given half the title (i.e., "Positive Impact of CryoSat-2 Ice Thickness Initialization") is not relevant to the Antarctic?

We added more discussion on Antarctica in Fig.1, in addition to Fig.3. We think there are some values in the discussion on Antarctica although we wish we could have more data to validate forecasts in the Antarctic.

Line 104: Perhaps similar mechanisms are at play here to the Arctic winter - i.e., ocean warming?

We have added this:

"It is possible that the missing mechanism in the ocean transport contributes to the RMSE in the Antarctic SIE."

Line 130: Can you say a little about how these SIE tendencies are calculated? I know that CICE includes tendencies for sea ice area caused by dynamics (daidtd) and thermodynamics (daidtt). Do you convert these to extent somehow? Or start from scratch?
Or are you really plotting the SIA trends (which would be more informative anyhow)?

Either way the caption/units are wrong because "cm/day" is only relevant for volume (or volume per unit area!).

Thanks for pointing out the wrong unit in Fig.7(c): the tendencies are measured in "%/day" for ice area and "cm/day" for ice volume per unit area. The area tendency from dynamics and thermo-dynamics (daidtd and daidtt) and the volume tendency from dynamics and thermo-dynamics (dvidtd and dvidtt) used here are directly from the model archives.

Line 133: I presume you are just plotting CICE's "meltt" here for top melting (and meltl & meltb for others)? If so you should consider adding the snow melt ("melts") to the plot or combining meltt & melts to plot a total top melting.
Because generally the ice doesn't start melting (i.e., meltt=0) until the snow is gone.

Exactly. We added the snow melting rate in addition to top, basal and lateral melting rate in Fig.7, all monthly mean and averaged north of 67.5ºN. Their variable names are melts, meltt, meltb and meltl, respectively from model archives. As expected, the snow melt peaked in June, while the rest peaked one month later in July.

Line 149: Are you basing that on the PIOMAS comparisons in Fig 7? If so the wording is a bit strong because PIOMAS is itself only a model. Also (as noted elsewhere) several studies have suggested that PIOMAS underestimates the volume of late winter Arctic sea ice when compared with CS2 radar altimetry (see Laxon et al., & Tilling et al. references from earlier comments).

(1) We added a sentence emphasizing that PIOMAS is a reanalysis product.
(2) As mentioned earlier, attached Fig.s3 shows that compared to CS2, PIOMAS often overestimates SIV in winter and spring and is comparable in fall. The CryoSat-2 values used here are consistent with Laxon et al. and the PIOMAS values used here are consistent with what is shown at the PIOMAS website.

Lines 155, 157 & 161: change "basin" to "region"
Done.

Line 165: CS2 errors in the BKG region might also be relatively higher? This area will contain a mixture of new ice alongside old/thick ice. In particular the Barents Sea is also impacted by storms and is the area of the Arctic where snow-ice formation occurs. Such conditions can cause problems for the CS2 observations because it impacts the radar penetration properties (i.e., radar freeboard vs true freeboard).

Thanks for the information.

Line 182: I struggle a bit here to see what the relevant mechanisms are here - partly because these processes are not really addressed in the text.
What are the mechanisms that might cause more surface/basal melting with thinner ice?

In many ways basal meting and surface melting are largely independent of ice thickness.

On the other hand it's quite obvious that if you have less ice then you might have lower melting fluxes - because it will require less melting (and in fact be able to take less melting) before it melts-out completely.

So what are the mechanisms here?
Is it the case that reduced ice concentration/cover is allowing more in-situ ocean warming (radiative and/or conductive), which will amplify the basal melting?

Or is this related to the temperature profile in the ice - thicker ice can support a larger thermal gradient, which needs to be warmed to freezing point before melting can occur?

The conductive heat flux at the bottom surface is inversely proportional to the ice thickness, as shown in Equation (72) on Page 33 of the CICE document. Unlike the basal melt, the top melt is not directly related to the ice thickness. Some surface properties like long wave emissivity and aerodynamic roughness are a function of the ice thickness. We have revised the text by focusing on the basal melting rate. We also added a plot of ice concentration to show that a lower ice concentration in the alt-init experiment could amplify the basal melt. As seen in the attached Fig.s5, in the region where the basal melt differs most, the ice thickness changes a lot and there is a difference in the ice area up to 15% in the southern part of this region as well. The revised text now says:

"With the same prescribed atmospheric forcings applied to both experiments as discussed in Section 2.1, a relatively thinner ice in the alt-init experiment favors a higher basal melt rate than in the control experiment during the month of July, suggesting an important role of a proper ice thickness plays in the basal melting process. In addition, a lower ice concentration up to 15% in the alt-init experiment would allow more in-situ ocean warming, which would amplify the basal melting."

Line 185: I strongly suspect it's one (or a combination) of the 1st two!

We added this:

"It is most likely due to the lack of interaction with the Atlantic Ocean, the mechanism that is missing in this experiment by using a simplified mixed layer ocean model mentioned earlier. Without northward oceanic heat transport from the Atlantic, the sea ice edge tends to extend too far southward in winter."

Line 191: Can you explain more? In many cases coupled feedbacks would be expected to increase RMSE and model drift!

Indeed, relaxation methods are often used in models to relax boundary conditions to observations in order to prevent models from drifting away. In this case, the prescribed atmospheric and ocean boundary conditions are model simulations, which can contain errors and thus lead to a bias in the sea ice model. When the error occurs in the sea ice model, it tends to stay in the model as there is no feedback mechanism in the uncoupled model to get rid of it.

We added this:

"this study uses an uncoupled sea ice model with prescribed atmospheric and ocean boundary conditions from model simulations, which may contain errors and are not always 'in tune' with the sea ice model."

[Figure]

Fig.s3: Arctic sea ice volume in CryoSat-2 (left) and the difference of PIOMAS and CryoSat-2 (right) from 2011 to 2017.

Sea Ice Edge (15% ice cover) Target Oct; IC Apr 1

[Figure]

Fig.s4: Sea ice edge (15% sea ice concentration) from control (blue) and alt-init (red) experiments in October 2013 - 2016, from models initialized on April 1 each year, and AMSR2

(black).

[Figure]

Fig.s5: Monthly mean Arctic SIA (%) and SIT (m) in the control (left), their differences (middle) and the melt difference (cm/day) at the top and basal (right) between the alt-init and control experiments. All valid in July 2016 initialized on January 1, 2016.